# Spatial-Temporal Characteristics and Driving Factors of the Eco-Efficiency of Tourist Hotels in China

**DOI:** 10.3390/ijerph191811515

**Published:** 2022-09-13

**Authors:** Duoxun Ba, Jing Zhang, Suocheng Dong, Bing Xia, Lin Mu

**Affiliations:** 1Tourism College, Northwest Normal University, Lanzhou 730070, China; 2Institute of Geographic Sciences and Natural Resources Research, Chinese Academy of Sciences, Beijing 100101, China; 3Management Institute of The People’s Republic of China, Ministry of Culture and Tourism Quality Supervision, Beijing 100740, China

**Keywords:** tourist hotel, eco-efficiency, carbon emissions, Super-SBM Non-Oriented, driving factors, China

## Abstract

At present, COVID-19 is seriously affecting the economic development of the hotel industry, and at the same time, the world is vigorously calling for “carbon emission mitigation”. Under these two factors, tourist hotels are in urgent need of effective tools to balance economic and social contributions with ecological and environmental impacts. Therefore, this paper takes Chinese tourist hotels as the research object and constructs a research framework for Chinese tourist hotels by constructing a Super-SBM Non-Oriented model. We measured the economic efficiency and eco-efficiency of Chinese tourist hotels from 2000 to 2019; explored spatial-temporal evolution patterns of their income, carbon emissions, eco-efficiency, and economic efficiency through spatial hotspot analysis and center of gravity analysis; and identified the spatial agglomeration characteristics of such hotels through the econometric panel Tobit model to identify the different driving factors inside and outside the tourist hotel system. The following results were obtained: (1) the eco-efficiency of China’s tourist hotels is higher than the economic efficiency, which is in line with the overall Kuznets curve theory, but the income and carbon emissions have not yet been decoupled; (2) most of China’s tourist hotels are crudely developed with much room for improving the economic efficiency, and most of the provincial and regional tourist hotels are at a low-income level, but the carbon emissions are still on the increase; and (3) income, labor, carbon emissions, waste emissions, and water consumption are the internal drivers of China’s tourist hotels, while industrial structure, urbanization rate, energy efficiency, and information technology are the external drivers of China’s tourist hotels. The research results provide a clear path for the reduction in carbon emissions and the improvement of the eco-efficiency of Chinese tourist hotels. Under the backdrop of global climate change and the post-COVID-19 era, the research framework and conclusions provide references for countries with new economies similar to China and countries that need to quickly restore the hotel industry.

## 1. Introduction

As the core sector of tourism, the hotel industry in the tourism sector has the largest economic growth and employment ratio. According to a report by the United Nations World Tourism Organization, the tourist hotel industry has driven 10.6% (334 million jobs) of employment worldwide [1]. At present, China is the country with the largest domestic tourist reception volume in the world, and the hotel industry plays a significant role in stimulating the economy. In 2019, the operating income of China’s tourist hotels accounted for 34.75% of the total tourism revenue [2]. However, the eco-environmental problems caused by tourist hotels are particularly significant in all sectors of the tourism industry. Researching the global tourism carbon footprint, tourism carbon emissions account for approximately 5% of the world’s total, of which 20% is caused by accommodation services, second only to tourism and transportation, and may further increase in the future [3]. As a developing country with the largest carbon emissions in the world, China’s carbon emissions caused by the continuous expansion of the tourism and hotel industry cannot be ignored [4]. The hotel industry is facing the dual contradiction of positive economic and social impacts and pressure on the ecological environment [5].

Balancing economic growth and the ecological environment in the hotel industry has become the focus of many countries and scholars, such as China [6], Thailand [7], and the European Union [8]. The existing research on the hotel industry mainly focuses on three parts. One is research on the ecological environment and carbon emissions [9,10]. Long, X.L. studied the environmental efficiency of the hotel industry in different regions of China from 2000 to 2013 and found that the carbon emissions of the hotel industry in China increased significantly [11]. The second study concerns the economic growth and economic efficiency of the hotel industry, including the research on the economic operation law and economic growth of the hotel industry [12], the total factor eco-efficiency, the comprehensive economic efficiency of hotel enterprises [13], and the influence of the hotel industry on regional economic growth and economic efficiency. Pieri believes that investment in new technology has brought high production efficiency and rapid economic growth to the hotel industry [14]. The third research includes the relationship between the economic growth of the hotel industry and the impact on the ecological environment [15]. At present, an increasing number of people are paying attention to the methods of reducing the impact on the ecological environment while ensuring economic development [11]. Some scholars have preliminarily tried to analyze the relationship between the impact of hotels on the ecological environment and economic growth. The research of Domingos et al. concluded that the panel data of hotels in European countries are in line with the Kuznets curve hypothesis [16]. Cha et al. studied the decoupling of income and carbon emissions of hotels [17]. In summary, the existing research has made many efforts to balance economic growth and environmental impacts in the hotel industry and has achieved good research results. The main conclusions are as follows: first, the tourist hotel industry is a sector with a great economic contribution and ecological environmental impact in the tourism sector; second, the tourist hotel industry still has great potential for improving economic efficiency and slowing the impact on the ecological environment; and third, the tourist hotel industry is influenced by regional development policies, capital, and industry. An important way to improve its economic efficiency and reduce its impact on the ecological environment is through integrating markets and policies. Fourth, for developing countries, the tourist hotel industry is mostly in its initial stage with rapid economic growth, and its impact on the ecological environment cannot be ignored. Therefore, an urgent problem for the lodging industry in developing countries in the post-epidemic era is discussing how to regain the economic growth rate and how to reduce the negative impact on the ecological environment, especially the carbon emission level, while the economy and society are developing positively. Further research is needed to provide more accurate optimization suggestions and countermeasures.

Facing the existing knowledge gap, this paper focuses on the following questions: how did the income, carbon emissions, economic efficiency, and eco-efficiency of Chinese tourist hotels in the past transformation and restructuring process evolve? What is the pattern of their space? Has economic income and carbon emissions been decoupled? What is the coupling relationship between economic efficiency and eco-efficiency? What are the driving factors and optimization paths to reduce the impact on the ecological environment while balancing economic growth in the future?

This research takes China’s tourist hotel industry as the research object against the background of rapid growth, institutional transformation, and ecological civilization construction. From the aspect of theoretical and practical contribution, this study combed the spatio-temporal evolution pattern of income, carbon emissions, eco-efficiency, and economic efficiency of Chinese tourist hotels in the past 20 years. The results aimed to narrow the spatio-temporal heterogeneity between the east and the west, give full play to the regional advantages of different regions, narrow the gap between the rich and the poor, and promote coordinated development in China. Moreover, this study constructed the evaluation index system and theoretical framework for measuring eco-efficiency; provided the idea for measuring the relationship between economic development and the ecological environment; regarded tourist hotels as an open eco-economic system interacting with the outside, thus constructing a two-stage regression model to explore the factors that directly and indirectly affect eco-efficiency; and provides a path to optimize eco-efficiency for business operators and government entities.

The rest of the paper is organized as follows: Section 2 constructs the research framework and discusses the research methods. The research framework includes the research boundary, road map, and research field; the research methods include models, indicators, and data sources. Section 3 introduces the research results and discussions, including the results and analysis. Section 4 introduces the implications and gives the conclusions.

## 2. Literature Review and Research Framework

### 2.1. Literature on the Carbon Emissions of Hotels

With the increasing global environmental crisis, society and academia have paid great attention to carbon emissions. Some scholars have evaluated the carbon emissions of scenic spots [18], tourist transportation [19,20], and tourist accommodation [19,21,22]. Because hotels are open 24 h a day, with a wide variety of functions and facilities and the different living habits of residents, they are considered to be the main source of carbon emissions after tourist traffic [23,24]. 

As early as the late 1980s, some hotels in Europe realized the role of hotels in environmental protection, gradually carried out environmental management, established their environmental standards, and achieved remarkable results [25,26]. In Britain in 2009, there was a study on measuring the carbon emissions generated when individuals or groups directly or indirectly consume certain products or services by using the life cycle assessment (LCA) [27]. However, Chinese scholars’ research on hotel carbon emissions is relatively late. China did not realize that low-carbon environmental protection would become a new trend in the hotel industry until 2010. Since then, different scholars have studied the low-carbon production [28], low-carbon consumption [6], low-carbon service [29], low-carbon management [30], and low-carbon image [31] of the hotel industry and achieved great results. However, there are few studies on the spatial pattern, evolution trend, and income relationship of carbon emissions of Chinese hotels over a long time series.

### 2.2. Literature on the Eco-Efficiency Evaluation of Hotels

The concept of eco-efficiency is to create more goods and services while using fewer resources and reducing their impact on the natural environment [32]. It focuses on the practice of the efficient use of resources for economic and environmental progress [33,34] and provides quantitative guidance for sustainable tourism growth [35,36].

Over the past few decades, many studies have employed DEA to evaluate tourism efficiency [13,37,38]. Since the hotel industry is a vital sector in tourism, there is quite an extensive amount of literature on hotel efficiency [39,40,41]. Johns et al. [42] were the first to compare hotels using DEA. Since then, many scholars have applied DEA models to evaluate the efficiency of the hotel industry in various regions, such as China [43], Korea [44], Ecuador [45], Taiwan [46], and so on.

Data envelopment analysis (DEA), as a non-parametric and total-factor analysis approach, is essentially a linear programming model that evaluates the efficiencies of decision-making units (DMUs) by calculating the best multiplier for inputs and outputs. It has been widely used in tourism eco-efficiency assessments [47,48], The simplest way to calculate eco-efficiency is to use the ratio between economic value and environmental load [49], while measurement results in this way are usually unsatisfactory. The traditional DEA method has limitations in measuring ecological efficiency. Therefore, this paper uses the Super-SBM Non-Oriented model with unexpected output to measure the eco-efficiency value of tourist hotels.

### 2.3. The Research Object

Tourist hotels in this paper refer to star-rated hotels. After the reform and opening up in 1978, China welcomed many foreign guests, and new hotels gradually developed all over the country. These hotels were called foreign-related hotels and star-rated hotels at that time [50]. China’s star-rated hotels refer to foreign-related tourist hotels that have been evaluated by the tourism administration department following GB/T14308 and obtained star ratings [51]. Beijing won the Olympic bid in 2001, and Shanghai won the World Expo bid in 2002, which put Beijing, Shanghai, and its surrounding cities into a period of large-scale hotel construction. In 2010, the Star Hotel Classification and Evaluation (GB/T14308-2010) was promulgated, and in 2013, the CPC Central Committee issued “eight regulations” requiring diligence and thrift, which led to a decline in the number of star-rated hotels in China. The number of star-rated hotels in China increased from 6029 in 2000 to 10,003 in 2019, as shown in Figure 1. During the research period of this paper, star-rated hotels in China have experienced the transformation from public ownership to private ownership and diversified mixed ownership, and the trajectory of economic development and its impact on the ecological environment were different from those of other countries, and they are representative [52]. In addition, the data of star-rated hotels are continuous, and the products and services of departments are independent. The calculation results accurately reflect the objective situation of the eco-economic system of tourist hotels. Since 2000, China’s tourist hotels have gradually developed into chains and groups, and after 2000, they have shown obvious industrial characteristics. In 2019, the COVID-19 epidemic broke out in an all-around way, causing almost devastating damage to the tourist accommodation industry, with discontinuous panel data and complex and changeable influencing factors.

### 2.4. The Research Framework

Based on the existing research, it is found that the single factor framework cannot accurately measure the eco-efficiency and economic efficiency, so in this paper, the empirical analysis mainly selects the following indicators according to the actual production process of hotels, including energy and water resources as the input indicators of the ecological environment; SO_2_, wastewater, domestic garbage, and carbon emissions are the unexpected output indicators of the ecological environment; labor and capital are the input indicators of economic efficiency, and income is the output indicator of economic efficiency. The abovementioned ecological environment and economic indicators are treated by the tourism stripping coefficient method. By constructing a Super-SBM Non-Oriented model including unexpected output, the eco-efficiency and economic efficiency of tourist hotels in 30 provinces and regions of the Chinese mainland are evaluated. By using spatial hotspot analysis and barycenter analysis, this paper discusses the moving track, temporal and spatial evolution patterns, and spatial agglomeration characteristics of the income, carbon emissions, eco-efficiency, and economic efficiency of tourist hotels in China. 

In this paper, the tourist hotel is no longer regarded as a closed system but assumed as an eco-economic system interacting with the outside of the hotel. Obviously, the external indicators of tourist hotels always affect eco-efficiency by influencing the internal indicators of tourist hotels instead of directly acting on tourist hotels. Based on this fact, this paper, through an econometric panel Tobit model, uses two-stage regression to find the internal and external factors that directly and indirectly affect the eco-efficiency of tourist hotels and puts forward targeted opinions and optimization paths.

The specific research framework is as follows (Figure 2):

Step 1: To build a Super-SBM Non-Oriented mode to measure the eco-efficiency and economic efficiency of tourist hotels and find the spatial distribution and temporal and spatial evolution pattern of income, carbon emissions, economic efficiency, and eco-efficiency of tourist hotels in China.

Step 2: To analyze the coupling relationship between income and carbon emission, economic efficiency, and eco-efficiency of Chinese tourist hotels through the CR model.

Step 3: To reveal the internal and external influencing factors of the eco-economic system of tourist hotels through the time-series Tobit regression model.

## 3. Methods and Data

### 3.1. Economic and Eco-Efficiency Analysis Methods: Super-SBM Non-Oriented

Since the traditional DEA model cannot solve the input–output slackness problem, the super-efficiency SBM model has been proposed based on the SBM model for the efficiency of effective decision units and is currently used in many fields [53]. In this study, the eco-efficiency and economic efficiency of tourist hotels are measured by constructing a Super-SBM Non-Oriented model that includes non-desired outputs, and each spatial unit includes three vectors of inputs, desired outputs, and non-desired outputs. Then, the Super-SBM Non-Oriented based on a variable payoff to scale is expressed as:(1)$$\rho =\left(\frac{1}{m}\overset{m}{\underset{i=1}{\sum }}\frac{x_{io}-s_{i}^{-}}{x_{io}}\right){\left(\frac{1}{s}\overset{s}{\underset{r=1}{\sum }}\frac{y_{ro}-s_{i}^{+}}{y_{ro}}\right)}^{-1}$$

Among them, ρ is the target efficiency value, namely the eco-efficiency value and economic efficiency value; *x* and y are input and output, respectively; and m and s are the numbers of input and output indicators. When ρ ≥ 1, the evaluated decision-making unit is relatively effective; when ρ < 1, the evaluated decision-making unit is relatively ineffective, so it is necessary to improve the input–output variables [54]. In this way, the levels of eco-efficiency and economic efficiency of tourist hotels in 30 provinces and regions in mainland China are judged. For the eco-efficiency of tourist hotels, this study classifies the eco-efficiency level of tourist hotels in each region into five levels from low to high and uses CE to express the eco-efficiency of tourist hotels, with 0.00 < CE < 0.40 as an inefficient region, 0.41 < CE < 0.60 as an inefficient region, 0.61 < CE < 0.80 as a region to be improved, 0.81 < CE < 1.0 as a relatively efficient area, and CE > 1 as a highly efficient area.

### 3.2. Spatial Pattern Analysis

#### 3.2.1. Hot Spot Analysis

The Getis-Ord *G_i_** spatial hotspot analysis, first proposed by Getis and Ord, can reflect the distribution of hot and cold spots in the local space of the study object [55]. It is expressed as:(2)$$G_{i}^{*}=\frac{\sum_{j=1}^{n}\varphi_{i,j}x_{j}-(\frac{1}{n}\sum_{j=1}^{n}x_{j})\times \sum_{j=1}^{n}\varphi_{i,j}}{\sqrt{\frac{\sum_{j=1}^{n}x_{j}}{n}-(\frac{1}{n}\sum_{j=1}^{n}x_{j}{)}^{2}}\times \sqrt{\frac{n\sum_{j=1}^{n}\varphi_{i,j}^{2}-{\left(\sum_{j=1}^{n}\varphi_{i,j}\right)}^{2}}{n-1}}}$$
where *x_j_* is the attribute value of element *j*, *φ_i_*_,*j*_ is the spatial weight between elements *i* and *j*, and *n* is the total number of elements. According to the *G_i_** index, it can show where spatially high value (hot spot) or low value (cold spot) elements occur clustered in space. The hot spots and cold spots of tourist hotel income, carbon emissions, eco-efficiency, and economic efficiency in 30 provinces and regions in mainland China represent the high-value significant areas of income and low-value significant areas, respectively.

#### 3.2.2. Analysis of the Center of Gravity Coordinates

The concept of the center of gravity comes from physics and geometry [56]. At present, some scholars have applied this model to the study of the economy or population. When the center of gravity of the regional economy is far from the geometric center of the region, it means that the regional economic development is in an unbalanced state, and the gap between the economic development levels of different regions is greater [57]. In this paper, barycenter coordinate analysis is used to analyze the moving track of the barycenter of national tourist hotels’ income, carbon emissions, eco-efficiency, and economic efficiency and to analyze the spatial evolution law of tourist hotels’ income, carbon emissions, eco-efficiency, and economic efficiency. Taking the calculation of eco-efficiency as an example, its mathematical expression is:(3)$$p_{i}\left(x_{i},y_{i}\right)=\left[\frac{\sum_{i=1}^{n}ce_{i}x_{i}}{\sum_{i=1}^{n}ce_{i}},\frac{\sum_{i=1}^{n}ce_{i}y_{i}}{\sum_{i=1}^{n}ce_{i}}\right]$$
where *p_i_* (*x_i_*, *y_i_*) denotes the center of gravity coordinates of eco-efficiency in region *i* and *ce_i_* denotes the eco-efficiency level of tourist hotels in region *i*.

### 3.3. CR Model: Coupling Relationship

Coupling refers to the phenomenon in which two or more systems influence each other and even unite through interaction [58]. The early coupling research was mainly applied in the engineering field and then gradually introduced into the research of social economy, resources and environment, economy and ecological environment quality, regional development intensity, and resources and environment as they relate to urbanization and ecological environment coupling [59]. Based on the coupling analysis of the income (I) and carbon emissions (C) of tourist hotels in China in 2000 and 2019, the results show that type I is of high economic development (I > 80, C > 10), type II is of high carbon and low-income imbalance (I < 80, C > 10), and type III is of extensive and low efficiency (I < 80, C < 10). Based on the coupling analysis of the economic efficiency (EE) and eco-efficiency (CE) of China’s tourist hotels in 2000 and 2019, the 30 provinces and regions in China are further divided into 4 types with different relative effective levels as the dividing line. Type I is high-carbon development (EE > 0.7, CE > 0.9), type II is low-carbon development (EE > 0.8, CE < 0.8), type III is extensive and inefficient (EE < 0.7, CE < 0.9), and type IV is low-carbon and inefficient (EE < 0.7, CE > 0.9)

### 3.4. Driver Analysis: Panel Tobit Analysis

The Tobit regression model was proposed by Tobin, and it focuses on the problem of model construction with restricted or truncated dependent variables [60]. The eco-efficiency and economic efficiency values in this study are in the range of (0, 1) or (0, +∞) intervals. For this situation where multiple samples converge to a certain limiting value within a specific range, the Tobit model can accurately explain the difference in nature between its limiting and nonlimiting observations [61]. The panel Tobit model was therefore used to identify the drivers of spatial variation affecting eco-efficiency in tourist hotels. The model expressions are as follows:(4)$$y_{it}^{\;*}=\alpha x_{it}+\varepsilon_{it}\;y_{it}=\left\{\begin{array}{c} y_{it}^{\;*},\;y_{it}^{\;*}\geq 0\;\\ 0,\;\;y_{it}^{\;*}\leq 0 \end{array}\;i=1,\cdots ,\;N\;and\;t=\right.1,\cdots ,T\;\varepsilon_{it}\sim N\left(0,\sigma^{2}\right)$$where *i* represents the 30 provinces and regions in mainland China, *t* represents different years, *x_it_* is the independent variable, *β* is the regression parameter, and $\varepsilon_{it}$ represents the disturbance term.

### 3.5. Indicators and Data

#### 3.5.1. Indicators of Economic and Eco-Efficiency Analysis

The measures of the economic efficiency of hotels include the investment, labor, and income of hotels. The measurement indices of hotel eco-efficiency include input, expected output, and unexpected output. There is no unified measurement index for the selection of input and output factors at home and abroad, at present [62,63]. Based on the actual input and output process of the development factors of the hotel industry, this paper selects specific measurement indices. The index units are shown in Table 1.

The input indicators of economic efficiency mainly include investment and energy consumption. Regarding output, the output value of the traditional economic system is selected as the measurement index, which is the income of the hotel industry.

The input indicators of eco-efficiency include labor, energy, investment, and water resources in the ecosystem. Among them, labor refers to the number of employees in tourist hotels, energy input refers to the calculation of the energy consumption level of tourist hotels through the consumption of standard coal per unit GDP of each industry in the regional energy balance table, and water resource input mainly refers to the water consumption of the urban population. Regarding output indicators, the income of tourist hotels is selected. Unexpected output indicators include wastewater discharge, garbage discharge, SO_2_ discharge, and carbon discharge. Wastewater discharge, garbage discharge, and SO_2_ discharge are the same as in the calculation method of water resource input. Carbon emissions are calculated using energy data according to IPCC international standards.

The investment and the income of tourism are based on the year 2000, and the fixed assets price index and consumer price index of each province and region are treated at the same price.

#### 3.5.2. Regression Model Indicator Selection and Model Building

The driving mechanism of the eco-efficiency of the hotel industry is to excavate the spatial heterogeneity factors of the eco-efficiency of the regional hotel industry [64]. In this paper, tourist hotels are regarded as a complete eco-economic system, and the indicators that directly affect the eco-efficiency value of tourist hotels in the system are the direct driving factors, while the indicators that affect the eco-efficiency value outside the system are regarded as the indirect driving factors. In this paper, nine indicators, including investment, labor, energy, WU, WW, garbage, SO_2_, CO_2_, and income, which directly affect the eco-efficiency of tourist hotels, are taken as the direct driving indicators in the tourist hotel system. Among them, the tourist hotel system mainly affects regional eco-efficiency through the environmental system, economic system, and resource system [65]. The environmental system includes garbage emissions, wastewater emissions, SO_2_ emissions, and CO_2_ emissions. The economic system mainly includes the hotel industry income representing the scale effect, the number of employees representing the labor effect, and the original value of fixed assets representing the capital effect. The system includes water consumption and energy input. The statistical description of the variables is shown in Table 2.

The expression of the panel Tobit first-stage regression model is as follows:(5)$$\begin{array}{l} EE_{it}=\alpha_{0}+\alpha_{1}lnIncome_{it}+\alpha_{2}lnCO_{2_{it}}\\ \;\;\;+\alpha_{3}{\mathrm{lninvestment}}_{it}+\alpha_{4}lnlabor+\alpha_{5}energy+\alpha_{6}lnwu+\alpha_{7}lnww+\alpha_{8}lngarbage+\alpha_{9}lnSO_{2}\;+\varepsilon_{it} \end{array}$$

Generally, its external driving force can be analyzed from the economic and social environment, which are closely related to the eco-economic system of tourist hotels [66,67], including the per capita GDP (GPC) representing the level of economic development, the proportion of tertiary industry (TI) representing the industrial structure, the proportion of the urban population (UP) representing the urbanization rate, and the per capita water resources (GPW) representing the level of resource utilization. The energy consumption per unit GDP (GPE) and SO_2_ energy consumption per unit GDP (GPSO_2_) represent regional energy efficiency; the number of ordinary college students (ST) represents civilization level; the highway mileage (KM) represents traffic level; the total investment of foreign enterprises at the end of the year (FI), the foreign exchange income of international tourism (FE), and the number of international tourists (IIT) represent the level of informatization; and the total post and telecommunications business per capita (PT).

According to the different regression results in the first stage, three systems and five regression models are formed in the second stage of this paper. Taking the income of tourist hotels as an example, the expression of the second-stage regression model is as follows:(6)$$\begin{array}{l} lnIncome=\alpha_{0}+\alpha_{1}lnGPC+\alpha_{2}lnTI+\alpha_{3}lnUP+\alpha_{4}lnGPW+\alpha_{5}lnGPE\\ \;\;\;\;\;+\alpha_{6}lnSO_{2}+\alpha_{7}lnST\\ \;\;\;\;\;+\alpha_{8}lnKM+\alpha_{9}lnFI+\alpha_{10}lnPT+\alpha_{11}lnFE+\alpha_{12}lnIIT+\varepsilon_{it} \end{array}$$

### 3.6. Data Sources

In this paper, the income, labor, and investment of tourist hotels are from the China Tourism Statistical Yearbook and its copies from 2001 to 2020. Per capita GDP, the proportion of urban population, the total post and telecommunications business, and GDP are from the China Statistical Yearbook and its copies. The total population and urban population are from the China Demographic Yearbook from 2001 to 2020. The price index data of the total investment of foreign enterprises at the end of the year and the number of international tourists received are from the National Bureau of Statistics of China; the data of per capita wastewater discharge, per capita garbage discharge, and per capita sulfur dioxide discharge are from the China Environmental Statistics Yearbook from 2000 to 2019; and the data of total energy, wholesale and retail hotel and catering energy consumption are from the China Energy Statistics Yearbook from 2001 to 2020.

## 4. Results 

### 4.1. General Trends

The development of tourist hotels in China is in accordance with the environmental Kuznets curve theory, but income and carbon emissions have not yet been decoupled. As shown in Figure 3, carbon emissions increase with the increase in the income of tourist hotels; after income reaches the inflection point, carbon emissions still show an increasing trend. There is an inverted U-shaped curve between the income of tourist hotels and carbon emissions, which is consistent with the environmental Kuznets curve theory [68]. From 2001 to 2012, the income growth rate of tourist hotels in China was above the original growth, which was positive growth. In 2013, it was at or below the original growth, which was negative growth, indicating that there was an inflection point in income in 2013. The growth rate of carbon emissions increased rapidly from 2000 to 2004 and exhibited a negative trend after 2008. During this period, China made many efforts in low-carbon development policies and environmental regulation, such as the promulgation of the Notice of the State Council on Printing and Distributing the Comprehensive Work Plan of Energy Conservation [69] and Emission Reduction in 2008 and the establishment of the concept of green emission reduction for hosting the Olympic Games [70]. In 2019, the average carbon emission growth rate reached its lowest value, indicating that China’s carbon emission reduction policy has achieved remarkable results. However, the highly consistent income growth rate and carbon emission growth rate indicate that the economic growth of tourist hotels and carbon emissions have not been decoupled.

In contrast to the regional eco-efficiency, the eco-efficiency of tourist hotels is higher than the economic efficiency. The eco-efficiency of tourist hotels in China fluctuates around the efficiency value of 1, with the highest eco-efficiency of 1.10 in 2000, followed by 1.04 in 2017, and the lowest in 2001. The eco-efficiency reached the peak value of 0.86 in 2012 and the lowest value of 0.54 in 2003, and the fluctuation of economic efficiency was greater than that of eco-efficiency. In the past 20 years, the economic efficiency of China’s tourist hotels has shown an increasing trend, while eco-efficiency has shown an insignificant increasing trend. 

### 4.2. Characteristics of the Spatial Distribution of Income and Carbon Emissions

In terms of spatial distribution, the income level of tourist hotels in eastern China is higher than that in western China, and the carbon emissions in western China are higher than those in eastern China. As shown in Figure 4, from 2000 to 2019, the income level of tourist hotels in China showed a high-income zone in the southeast coastal areas, a middle-income zone in the middle, and a low-income zone in the west and north. From 2000 to 2019, the spatial distribution of China’s carbon emissions showed that the northern and western regions were high-carbon emission zones, the central region was a medium-carbon emission zone, and the southeast region was a low-carbon emission zone. According to the carbon emission level and income level of tourist hotels in China in 2000, 2009, 2015, and 2019, the carbon emissions of some western provinces and cities continued to increase with the increase in tourist hotel income, and the carbon emissions in the southeast coastal areas showed a downwards trend after the income reached a certain level. This further proves that the income and carbon emissions of tourist hotels conform to the environment Kuznets curve. Most provinces and regions in the west with extensive development as the main mode are in the first half of the Kuznets curve, and some eastern provinces have entered the second half of the Kuznets curve and are gradually realizing weak decoupling. However, the weak decoupling state can only show that economic growth depends on environmental pollution [71].

The income of China’s tourist hotels has an obvious spatial spillover effect, and carbon emissions have not formed spatial diffusion. The hot spot of hotel income level is located in the Yangtze River Delta region of China and has formed a good spatial spillover effect, which is consistent with previous studies [72]. The cold spots of the carbon emissions of tourist hotels in China are not obvious, which indicates that the low-carbon emissions of tourist hotels in China have not formed spatial diffusion. Carbon emission hotspots shifted from the east to the west from 2009 to 2019. Combined with the spatial distribution of hot spots in tourist hotels, China’s eastern coastal cities are gradually decoupling income from carbon emissions.

The focus on the carbon emissions and income level of Chinese tourist hotels has shifted from southeast to northwest. Given the changing track of the income center of gravity, the income level in the western region has increased in the past 20 years, which has a strong traction effect on the center of gravity, making the overall track of tourist hotels show a shock shift from southeast to northwest. Based on the changing track of the carbon emission center of gravity, the overall track shows a shock shift from southeast to northwest, and the east–west amplitude is greater than the north–south amplitude, which indicates that the difference in carbon emissions between east and west tourist hotels is greater than that between north and south. The strong traction force in the west and the reduced traction force in the east shift the center of gravity from east to west. Therefore, the economic growth of tourist hotels in western China failed to coordinate the relationship between economic development and the ecological environment, resulting in a large amount of carbon emission. 

### 4.3. Characteristics of the Spatial Distribution of Economic Efficiency and Eco-Efficiency

Economic efficiency has no obvious agglomeration characteristics in space, but eco-efficiency has obvious regional polarization. As shown in Figure 5, from 2000 to 2019, the spatial distribution of the economic efficiency and eco-efficiency of tourist hotels in China showed that the eastern and western regions were high-efficiency zones, the central region was a medium-efficiency zone, and the northeast region was a low-efficiency zone. As far as economic efficiency is concerned, China’s five-star hotels are mainly distributed in the southeast coastal areas; under the current situation of China’s development, the economic efficiency of high-star hotels is low Therefore, even in the southeast coastal areas with obvious income agglomeration characteristics, there are no agglomeration characteristics of economic efficiency. The eco-efficiency shows obvious regional polarization, with Sichuan in the west, Jilin in the northeast, Ningxia in the middle, and Zhejiang in the east. The institutional constraints of these four provinces are at the forefront. For example, Zhejiang put forward the criteria for selecting green hotels as early as 1999 and took the lead in creating green consciousness [67].

The eco-efficiency and economic efficiency of Chinese tourist hotels have no good spatial spillover effect. The provinces with hot economic efficiency are not obvious, which indicates that the provinces with the better economic efficiency of tourist hotels have not realized the spillover effect to the surrounding areas. The eco-efficiency of tourist hotels is a transitional area with insignificant correlation, which indicates that the spatial aggregation effect of eco-efficiency of tourist hotels in China is not significant.

The center of economic efficiency shifts from east to west, and the center of eco-efficiency shifts from west to east. Based on the trajectory of economic efficiency, in recent years, the center of the economic efficiency of tourist hotels in China has shifted westward. The improvement of transportation and production technology in western China shows that accessibility and high production efficiency can improve the economic efficiency of tourist hotels [73]. Based on the changing track of the eco-efficiency center of gravity, the overall track shows a shock shift from west to northeast, and the shock range from east to west is greater than that from south to north, which indicates that the eco-efficiency difference between tourist hotels is greater than the north–south difference. Even though the western region has provinces with a good ecological background, its traction on the eco-efficiency center is not strong, which shows that relying on the environmental background cannot achieve long-term eco-efficiency advantages; only steadily developing the tourist hotel economy and minimizing the environmental cost are the most effective ways to achieve high eco-efficiency.

### 4.4. Coupling Analysis of Income and Carbon Emissions, Economic Efficiency, and Eco-Efficiency

The income (I) and carbon emissions (C) of tourist hotels in China in 2000 and 2019 are coupled, and the economic efficiency (EE) and eco-efficiency (CE) are coupled. As shown in Figure 6, the scatter analysis of the above two groups of variables shows that there is a linear positive correlation. The contribution rate of unit income to carbon dioxide emissions in 2000 is 0.19 and that of unit eco-efficiency to economic efficiency in 2019 is 0.79. China’s 30 provinces and regions are divided into four types by using different relative effective levels as dividing lines, as shown in Table 3.

Most of China’s tourist hotels are roughly developed and tend to increase their carbon emissions despite their low-income levels. In the coupling relationship between carbon emissions and income of China’s tourist hotels, provinces in Type III regions steadily transition to Type I and Type IV regions, and most of them have a continuous trend of transitioning to Type II regions. Only Jiangsu Province, a Type IV region, has made the direct leap from a Type III to a Type IV region. Guangdong, Zhejiang, and Sichuan have achieved a direct transition from low carbon emission and low-income levels to high-carbon emission and high-income levels. In the future, most provinces in China should focus on carbon reduction while maintaining steady income growth.

The economic efficiency of China’s tourist hotels has great room for improvement, and the eco-efficiency changes slowly. In the coupling relationship between the economic efficiency and eco-efficiency of tourist hotels in China, most provinces are distributed in type III and type IV regions, and the eco-efficiency level of some provinces is still declining. The number of provinces in the Type I area is small. Ningxia, as a province with a better eco-environmental background, has achieved good results by using the hotel industry to promote economic development. In the future, Ningxia should exert a spillover effect to drive the economic efficiency and eco-efficiency of underdeveloped areas in the west.

### 4.5. Study of Driving Factors

#### 4.5.1. First Stage Regression

The investment and income of hotels are based on the year 2000, and the fixed assets price index and consumer price index of all provinces are used for constant price treatment. To avoid the nonstationarity of parameter estimation caused by different data dimensions, the natural logarithm of related variables is taken to preserve the characteristics of panel data [74]. The regression results are shown in Table 4.

The regression results revealed the following:

In the environmental system, the most significant indicator of eco-efficiency is carbon emissions, followed by garbage emissions. For every 1% increase in carbon emissions, the eco-efficiency of tourist hotels will decrease by 0.30%, and for every 1% increase in garbage emissions, it will decrease by 0.14%. In the resource system, the most significant indicator of eco-efficiency is the water consumption of tourist hotels. Every 1% increase in water consumption will lead to a decrease in eco-efficiency of 0.13%. In the economic system, the most significant indicator of eco-efficiency is the income of tourist hotels, representing the scale effect, followed by the number of employees in the hotel industry, representing the labor effect. Every 1% increase in the income of tourist hotels will increase eco-efficiency by 0.45%, and every 1% increase in labor input will lead to a decrease in eco-efficiency by 0.22%.

Within China’s tourist hotel system, income, labor, carbon emissions, garbage emissions, and water consumption directly affect the eco-efficiency level of tourist hotels. Through the first stage of regression, we find that in the environmental system, the carbon emissions and garbage emissions of China’s hotel industry harm its eco-efficiency. As a suggestion, by optimizing the “three wastes” emission system, green and low-carbon development will become the inevitable trend in global hotel industry development. In the economic system, the income representing the scale effect of tourist hotels has the most significant impact on eco-efficiency. This is because China’s tourist hotel industry has adjusted its industrial structure earlier than tourist attractions and travel agencies, and the new industrial structure that drives the development of the hotel industry for profit leads the income level to significantly improve eco-efficiency. In the resource system, water consumption has a great influence, so the idea of energy savings and emission reduction should be advocated to improve the recycling rate.

#### 4.5.2. Second Stage Regression

According to the first-stage regression, the second-stage regression selects the indicators that have a great influence on the eco-efficiency of tourist hotels in the economic system, environmental system, and resource system; that is, the income and labor of tourist hotels in the economic system, CO_2_ emissions and garbage emissions in the environmental system, and water consumption in the resource system, with a focus on analyzing the influence degree of external factors of the tourist hotel system on internal factors and establishing five models. The regression results are shown in Table 5.

The regression results show that the proportion of tertiary industry has a significant impact on the economic system, environmental system, and resource system. In addition to the influence of the proportion of the tertiary industry, in the economic system, the indicators that have a great influence on income and labor are the proportion of the urban population and the total post and telecommunications business per capita. For every 1% increase in the proportion of the urban population, the labor force decreases by 0.43%, and for every 1% increase in the total post and telecommunications business per capita, the income increases by 0.44%. In the environmental system, the indicators that have a great impact on carbon emissions and garbage emissions are the proportion of the urban population and per capita GDP. For every 1% increase in per capita GDP, garbage emissions will increase by 0.60%. In the resource system, the indicator that has a great impact on water consumption is energy consumption per unit of GDP. For every 1% increase in energy consumption per unit of GDP, water consumption will decrease by 0.59%.

Outside of China’s tourist hotel system, the industrial structure, urbanization rate, energy efficiency, and informatization degree indirectly affect the eco-efficiency level of tourist hotels. Through the second-stage regression, it is found that the per capita GDP, which represents the level of economic development, has no significant impact on the economic system, indicating that China’s rapidly growing regional economic level has not had a positive impact on the economic efficiency of tourist hotels during the study period. The proportion of the tertiary industry, which represents the industrial structure, has a positive impact on the economic system and resource system of tourist hotels, indicating that the eco-efficiency level of tourist hotels can be promoted by optimizing the proportion of tertiary industry. The rational allocation of the regional energy consumption structure and the improvement of informatization levels can indirectly promote the eco-efficiency level of tourist hotels. The degree of energy efficiency and informatization affects the eco-efficiency of tourist hotels by affecting the resource system and economic system.

## 5. Discussions

The focus of income and carbon emission of tourist hotels in China has shifted from the southeast to the northwest, which shows that the western provinces have not coordinated the relationship with the ecological environment while developing their economy rapidly. According to previous studies, economic development at the cost of the environment is not the best development path [75]. It is suggested that the western region of China should adjust its development strategy in time and emphasize the irreversibility of environmental damage [76]. In this study period, the eco-efficiency of Chinese tourist hotels is generally higher than the economic efficiency, which is different from the general law of the regional eco-efficiency of Chinese provinces. Xia studied the regional economic efficiency and eco-efficiency of tourist hotels in China in 2016 and found that the economic efficiency of each province is higher than the eco-efficiency level [13]. According to the existing research, it is found that the eco-efficiency of regional tourism under the constraint of the ecological environment shows an obvious downwards trend [77], while a slight increase in the eco-efficiency of tourist hotels shows that, compared with other tourism sectors, Chinese tourist hotels have comparative advantages in balancing economic growth and ecological environmental impact. These findings will facilitate researchers’ focus on carbon reduction measures in western provinces.

From this study, it is found that the development of China’s tourist hotels is still in the extensive development stage. It has long been found that the technical efficiency of China’s tourist hotels is low when studying the space–time heterogeneity of the hotel industry’s environmental efficiency [11]. For example, operators refuse to use solar energy instead of electric energy because they are worried that unstable solar energy will be complained about by customers on rainy, cloudy, and snowy days, thus affecting their operating income. Tang found that there are obvious phenomena of food waste and water waste in China’s high-star hotels [50]. Jie studied the low-carbon behavior of employees in star-rated hotels and found that the employees in star-rated hotels had insufficient low-carbon operation concepts, which could not guide consumers well [78]. Combined with the research in this paper, it is found that if tourist hotels cannot adjust their business philosophy and structure in time, they will still be unable to realize the green and low-carbon transformation for a long time to come.

The internal and external indicators of the star-rated hotel system directly and indirectly affect eco-efficiency. Moutinho and Robaina-Alves studied tourism satellite accounts in Portugal by using decomposition analysis [79,80]. Both studies found that the tourism activity effect, energy over fixed capital, and capital over labor productivity all have significant impacts on the accommodation industry. When studying the eco-efficiency of tourism [5], it is found that energy technology is the key to improving the eco-efficiency, which is consistent with the internal research of the star-rated hotel system in this paper; that is, excessive energy consumption and capital investment have a negative impact on the eco-efficiency of star-rated hotels. At the same time, according to the input–output process of star-rated hotels, some factors often do not directly affect the eco-efficiency of hotels. Li found that the optimization of industrial structure and urbanization rationalized the allocation of labor and energy consumption in the hotel industry, and the improvement of industrial structure would also improve the efficiency of CO_2_ emission [81]. In addition, with the increase in the proportion of urban residents, hotels could reduce the idle labor force. This is the same as the research result of this paper. These findings are helpful for managers to know how to improve eco-efficiency.

## 6. Conclusions and Implications

### 6.1. Conclusions

The eco-efficiency of China’s tourist hotels is higher than the economic efficiency, which generally conforms to the Environmental Kuznets curve theory. In terms of spatial distribution, the income level of China’s tourist hotels in the east is higher than that in the west, the carbon emissions in the west are higher than that in the east, the economic efficiency has no obvious agglomeration characteristics, and the eco-efficiency has a regional polarization effect, but it does not have a spatial spillover effect. On the shift of the center of gravity, carbon emissions and income shift from southeast to northwest, economic efficiency shifts from east to west, and eco-efficiency shifts from west to east. We should give full play to the linkage effect between the areas with good development of tourist hotels and the surrounding provinces.

Most of China’s tourist hotels are roughly developed, and most provinces and regions still have an increasing trend of carbon emissions at the low-income level. The economic efficiency of China’s tourist hotels has much room for improvement. In the future, China’s tourist hotels should speed up the transition from a rough development mode to a low-carbon sustainable development mode; promote Jiangsu Province’s exemplary role in the development of the national tourist hotel industry and Ningxia’s exemplary role in underdeveloped areas such as the northwest and southwest.

Inside China’s tourist hotel system, income, labor, carbon emissions, garbage emissions and water consumption in the economic, environmental, and resource systems directly affect the eco-efficiency of tourist hotels. Outside of China’s tourist hotel system, the proportion of the tertiary industry, the proportion of the urban population, energy consumption per unit of GDP, and total post- and telecommunications business per capita indirectly affect the eco-efficiency of tourist hotels. Therefore, within the hotel industry, the scale effect and capital effect need to be optimized, and the emission system needs to be adjusted. Regional governments should optimize the industrial structure of the hotel industry and improve the urbanization rate, energy efficiency, and information technology.

### 6.2. Implications

The theoretical significance of this study is constructing a research framework from measurement to pattern analysis to driving mechanisms, attempting to find the scientific tools and paths for the green and low-carbon sustainable development of tourist hotels. This study uses the Super-SBM Non-Oriented model, constructs the index system and theoretical framework for measuring eco-efficiency firstly, and reveals the spatio-temporal evolution pattern of the income, carbon emissions, eco-efficiency, and economic efficiency of Chinese tourist hotels from 2000 to 2019. It is hoped that the research results can provide ideas for balancing economic development and the ecological environment, as well as suggestions for regional coordinated development in China. Secondly, through the CR model, the coupling relationship between income and carbon emission, eco-efficiency, and economic efficiency is found. It is hoped that the research results can provide targeted transformation strategies according to the development characteristics of the hotel industry in different provinces. Finally, this paper regards tourist hotels as a complete eco-economic system, and through a two-stage regression model, investigates the factors that directly and indirectly affect the eco-efficiency of tourist hotels, hoping that the research results can provide ideas for enterprises and governments to improve the eco-efficiency. Moreover, the following recommendations are made based on the results of this paper’s empirical study:

Consolidate and expand the advantages of the Yangtze River Delta region, strengthen the radiation of the Pearl River Delta region to the southwest region, and promote the linkage effect of the Beijing–Tianjin–Hebei region to the northeast and northwest regions. Provinces with insignificant spillover effects should take the initiative to help neighboring provinces and cities and establish a cooperation mechanism for inter-provincial tourist hotels. For example, holding a sharing meeting of hotel management experience, establishing a sharing discount mechanism, and exchanging customer sources based on ensuring customer privacy. The provinces without development advantages should actively learn from the advanced technology and management experience of tourist hotels in neighboring provinces. For example, referring to the development history of excellent hotels and combining their actual conditions, they should formulate development strategies, ask relevant industry experts for technical guidance, and train their employees regularly [82].

Accelerate the transition of China’s tourist hotels from a crude development model to a low-carbon, green, and sustainable development model. The provinces with no advantages in eco-efficiency should deepen the reform of the property rights system of the hotel industry based on their resource advantages, promote the diversification of the property rights structure of hotels, learn from the development experience of the hotel industry in Jiangsu Province, and realize the transformation from a low-carbon emission economic province to a high-carbon emission economic province as soon as possible. For example, in the production process, the geothermal system is used instead of gas heating, a heating and air conditioning system that can be turned off automatically when guests leave the room to sleep and LED energy-saving lamps are used, heat insulation window materials to prevent heat loss are used, and employees are encouraged to use environmentally friendly vehicles [83]. In the consumption process, ceramic tableware is used instead of plastic tableware, and slogans with moral constraints are posted, such as “saving resources is everyone’s responsibility” and “protecting the earth, refusing to waste”. Long-term customers change bed sheets and toiletries as required instead of daily changes [84], rainwater is collected and purified for watering flowers and raising fish [84], and consumers are encouraged to use non-disposable products [78].

Enterprises and the government should jointly adjust the internal and external influencing factors of the tourist hotel system, and then adjust the eco-efficiency of tourist hotels. Within the tourist hotel system, enterprises can directly adjust the eco-efficiency of tourist hotels by adjusting the capital effect, scale effect, water consumption, and carbon emission, such as limiting the proportion of technical investment in the initial investment and replacement investment in the hotel industry [85] and using short videos to publicize and expand the market [86]. Outside the tourist hotel system, enterprises should call on the government to adjust the local industrial structure, urbanization, energy efficiency, and information degree as soon as possible to indirectly adjust the eco-efficiency of tourist hotels. For example, to expand the scale of hotel construction to increase the proportion of tourism and other tertiary industries [87], promote exchanges between communities and tourist hotels to raise residents’ awareness of environmental protection [88], publicize the awareness of energy efficient use and regularly assess the results [89], and improve the degree of information construction and transportation convenience [90].

### 6.3. Limitations and Further Research

This study takes tourist hotels in China as an example of a new economy country. There are huge differences in natural environmental conditions and tourism resource endowments among the 30 provinces in China. However, most of the driving factors in this paper remain at the economic and social levels, without taking into account the natural geographical factors and environmental factors. At present, this is the limitation of this paper, and it is also the focus of future research. In future research regarding the selection of indicators, more technical means should be used to obtain more geographical environmental indicators to explore the multidimensional driving factors, and the results should focus on the accuracy of geographical space and the pertinence of policy recommendations. This paper aims to provide a reference for other new economies and countries that are similar to China.

## Figures and Tables

**Figure 1 ijerph-19-11515-f001:**
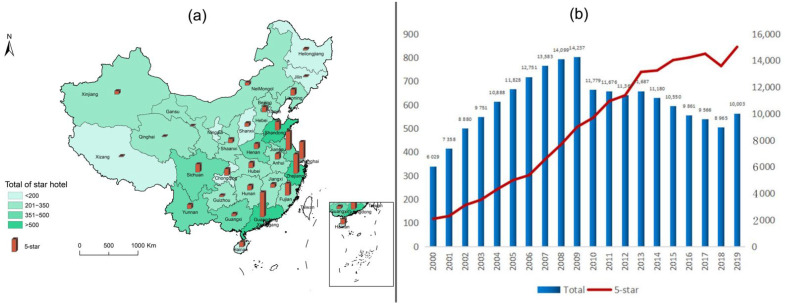
The spatial distribution pattern and quantity change of star hotels and five-star hotels in China. (**a**) number of star hotels and 5-star in 2019; (**b**) number of star hotels and 5-star in 2000–2019).

**Figure 2 ijerph-19-11515-f002:**
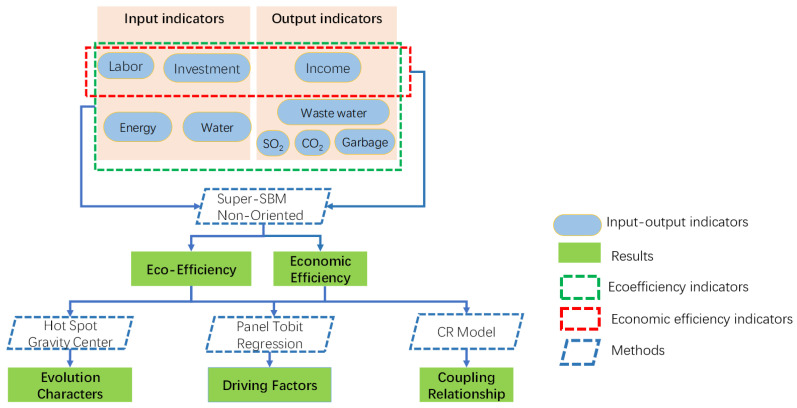
The analysis method of efficiency and eco-efficiency: Super-SBM Non-Oriented.

**Figure 3 ijerph-19-11515-f003:**
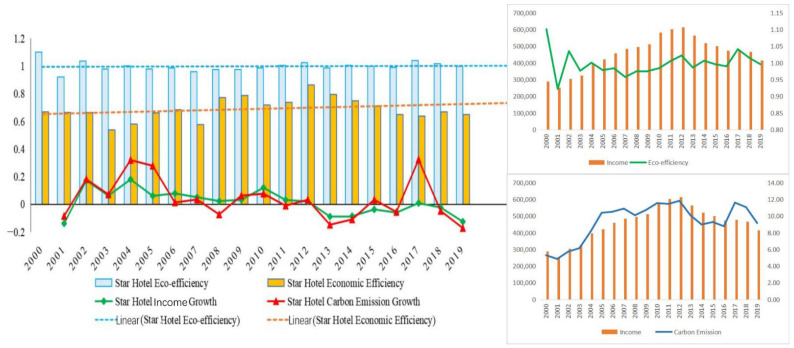
Overall trends in the eco-efficiency and economic efficiency of tourist hotels, 2000–2019.

**Figure 4 ijerph-19-11515-f004:**
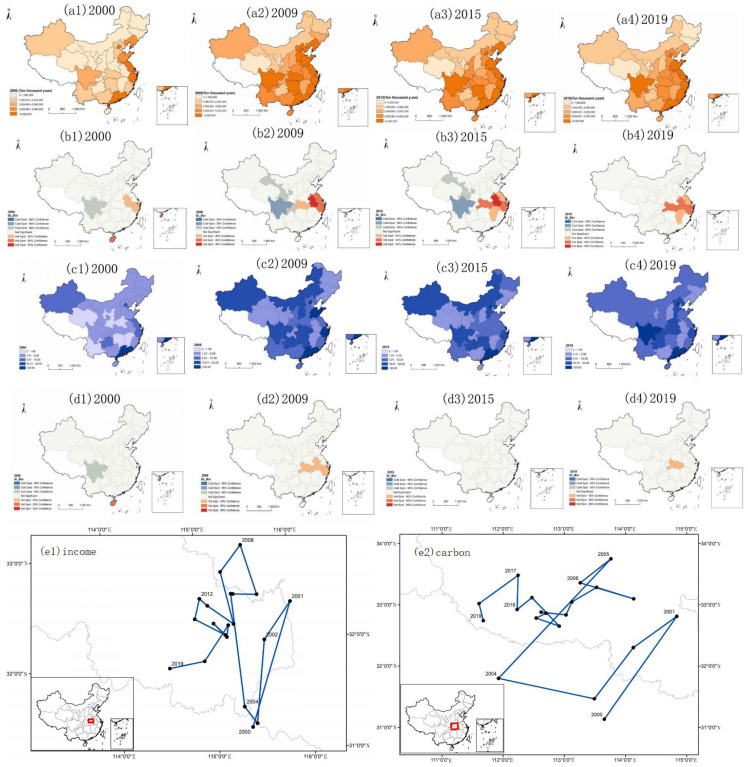
Spatial distribution characteristics and evolutionary trajectory of tourist hotel income and carbon emissions in China (**a1**–**a4**) is the spatial distribution chart of income in 2000, 2009, 2015, and 2019. The darker the color, the higher the income; (**b1**–**b4**) is the hot spot distribution of tourist hotel income; (**c1**–**c4**) is the spatial classification chart of carbon emissions, and the darker the color, the more serious the carbon emissions are. (**d1**–**d4**) are hot spots of carbon emissions in tourist hotels; (**e1**) is the shift track of the income center of China’s tourist hotels from 2000 to 2019, and (**e2**) is the shift track of the carbon emission center of China’s tourist hotels from 2000 to 2019. The red square indicates the position of the center of gravity in China.

**Figure 5 ijerph-19-11515-f005:**
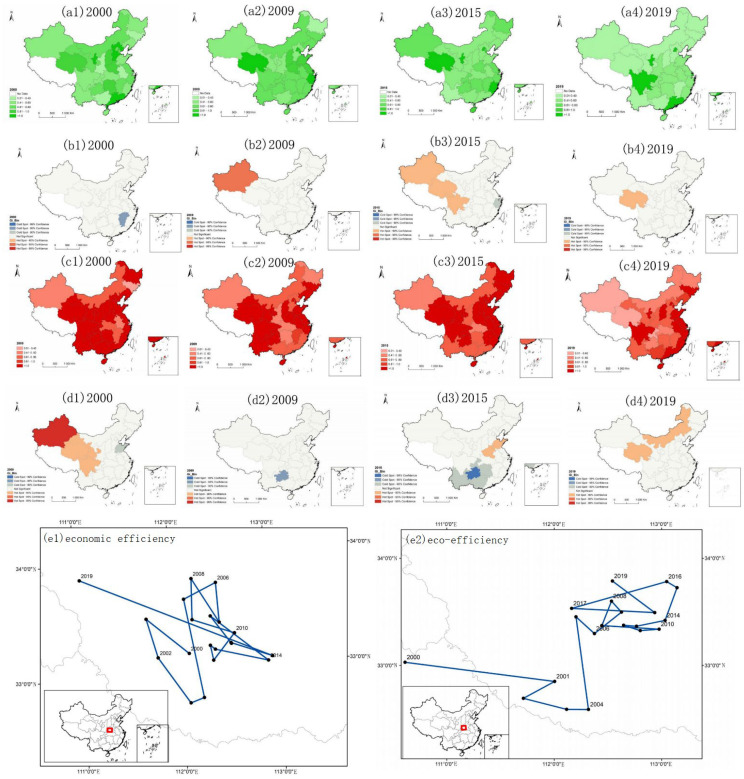
Characteristics and evolutionary trajectory of the spatial distribution of eco-efficiency and economic efficiency of tourist hotels, 2000–2019 (**a1**–**a4**) are charts of economic efficiency levels in 2000, 2009, 2015, and 2019. The darker the color, the higher the economic efficiency; (**b1**–**b4**) are the hot spots of economic efficiency of tourist hotels; (**c1**–**c4**) are the eco-efficiency level maps. (**d1**–**d4**) are the hot spots of the eco-efficiency of tourist hotels; (**e1**) is the shifting track of the economic efficiency center of China’s tourist hotels from 2000 to 2019, and (**e2**) is the shifting track of the eco-efficiency center of China’s tourist hotels from 2000 to 2019.The red square indicates the position of the center of gravity in China.

**Figure 6 ijerph-19-11515-f006:**
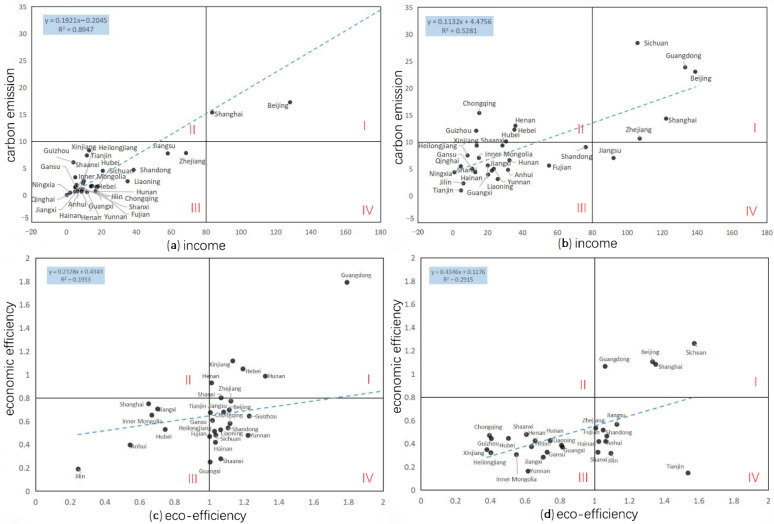
Coupled analysis of income–carbon emissions and economic efficiency–eco-efficiency of China’s tourist hotels in 2000 and 2019 (Qinghai is the heterogeneous point in (**c**) and Ningxia is the heterogeneous point in (**d**)). (**a**) Coupled income and carbon emissions in 2000; (**b**) coupled income and carbon emissions in 2019; (**c**) coupling of economic efficiency and eco-efficiency in 2000; (**d**) Coupling of economic efficiency and eco-efficiency in 2019.

**Table 1 ijerph-19-11515-t001:** Economic efficiency and eco-efficiency input–output indicators for tourist hotels.

		Indicators	Unit
Economic efficiency	Input	Investment	Million
		Energy consumption	Million tonnes of standard quasi-coal
	Output	Income	Million
Eco-efficiency	Input	Labor	10 k people
		Investment	Million
		Energy consumption	Million tonnes of standard coal
		Water	Million tonnes
	Output	Income	Million
	Undesirable output	Wastewater discharge	Million tonnes
		Garbage emissions	Million tonnes
		SO_2_ Emissions	Ton
		Carbon emissions	Ton

**Table 2 ijerph-19-11515-t002:** Statistical characteristics of driving variables.

VarName	Mean	SD	Min	Median	Max
EE	0.9976	0.3530	0.1552	1.0335	5.2535
lnIncome	12.5221	1.0435	7.6629	12.5215	14.8061
lnCO_2_	1.8171	0.9626	−2.7498	1.8077	4.1505
lninvestment	13.4911	0.9234	10.1831	13.4518	15.5445
lnlabor	10.4480	0.7903	7.2442	10.4832	12.2414
energy	12.3533	12.0678	0.0846	8.0623	83.9339
lnWU	14.5135	1.0303	11.4487	14.6088	16.7044
lnWW	13.8556	1.1297	10.5232	13.7814	16.4624
lngarbage	8.4608	1.0221	5.4709	8.3801	11.2042
lnSO_2_	6.1423	1.0271	2.1675	6.2885	7.8605

**Table 3 ijerph-19-11515-t003:** Measures taken by different types of regions and changes in the number of provinces.

Coupling Relationships	Type	Features	Measures	Number in 2000	Number of 2019
Income and CO_2_	I	High income, high carbon emissions	Develop a low carbon emission reduction and green development strategy.	2	5
	II	Low income, high carbon emissions	Restructuring and scaling of industries.	0	5
	III	Low income, low carbon emissions	Optimize the allocation of resources and give full play to the advantages of resources.	28	19
	IV	Low carbon emissions, high income	Enhancing spillover effects.	0	1
Economic efficiency and eco-efficiency	I	High economic efficiency, high eco-efficiency	To drive the development of neighboring provinces, and achieve green and low-carbon sustainable development of tourist hotels.	6	4
	II	High economic efficiency, low eco-efficiency	Adjusting energy allocation and developing energy-saving and emission reduction plans to improve clean production and green services.	0	0
	III	Low economic efficiency, low eco-efficiency	Improve the efficiency of capital and the quality of human capital.	7	12
	IV	Low economic efficiency, high eco-efficiency	Develop the tourist hotel industry, respond to China’s energy-saving and emission reduction policy, and learn from the production and operation mode of the I-type area.	17	8

**Table 4 ijerph-19-11515-t004:** Tobit regression results for internal driving forces controlling regional tourist-hotel eco-efficiency.

EE	Conf.	Z
lnIncome	0.447	9.027 ***
lnCO_2_	−0.300	−10.419 ***
lninvestment	−0.114	−2.365 **
lnlabor	−0.223	−4.264 ***
lnWU	−0.126	−3.834 ***
lnWW	0.075	1.857 *
energy	0.022	11.845 ***
lngarbage	−0.144	−3.744 ***
lnSO_2_	−0.045	−2.647 ***
_cons	1.812	7.050 ***
N	600	

Note: *, **, and *** were shown to be significant at the 0.1, 0.05, and 0.01 levels.

**Table 5 ijerph-19-11515-t005:** Tobit regression results for external driving forces controlling regional tourist-hotel eco-efficiency.

	Economic System				Environmental Systems				Resource System	
	Model I		Model II		Model I		Model II		Model II	
	lnIncome		Labor		CO_2_		lngarbage		lnWU	
	Coef.	Z	Coef.	Z	Coef.	Z	Coef.	Z	Coef.	Z
lnGPC	0.207	2.439 **	−0.007	−0.102	0.363	3.077 ***	0.603	6.211 ***	−0.149	−1.6
lnTI	1.535	9.597 ***	1.151	8.635 ***	1.69	7.279 ***	1.741	9.599 ***	1.506	8.545 ***
lnUP	−0.601	−6.465 ***	−0.433	−5.580 ***	−0.778	−5.748 ***	−0.378	−3.583 ***	−0.57	−5.569 ***
lnGPE	−0.169	−2.656 ***	−0.067	−1.263	0.483	5.131 ***	0.032	−0.448	−0.59	−8.365 ***
lnST	0.205	5.033 ***	0.218	6.437 ***	0.208	3.543 ***	0.259	5.604 ***	−0.116	−2.586 ***
lnKM	0.209	4.876 ***	0.285	8.006 ***	0.156	2.414 **	0.279	5.738 ***	0.332	6.968 ***
lnFI	0.236	7.460 ***	0.17	6.452 ***	−0.121	−2.645 ***	0.063	1.759 *	0.154	4.448 ***
lnPT	0.415	5.787 ***	0.359	6.004 ***	0.696	6.516 ***	0.646	7.942 ***	0.45	5.753 ***
lnIIT	0.156	5.845 ***	0.131	5.906 ***	0.312	7.740 ***	0.104	3.433 ***	0.297	10.067 ***

Note: *, **, and *** were shown to be significant at the 0.1, 0.05, and 0.01 levels.

## Data Availability

The data presented in this study are available in the Yearbook of China Tourism Statistics, which can be found at: https://data.cnki.net/yearbook/Single/N2020030028 (accessed on 30 July 2022).

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
