# Peer review of "Spatial-Temporal Characteristics and Driving Factors of the Eco-Efficiency of Tourist Hotels in China"

_ijerph, 2022, doi:10.3390/ijerph191811515_

Round 1

Reviewer 1 Report

The arrangement of sentence placement must be re-arranged, according to the existing template, because there is still a lot of empty space on the left side.

the purpose of this research for the contribution has not clear, 

the suggestions from the conclusions are also still hanging, please provide realistic suggestions so that business people can also get a direct impact, noy just ". Regional governments should optimize the industrial structure of the hotel industry and improve the urbanization rate, energy efficiency, and information technology." 

for example : "improve the urbanization rate" do you have provided the data? is this still relevant to this research

overall your research is good and very interested  

Author Response

Dear Reviewer,

Thank you very much for your professional and crucial comments and guidance, and we do think it is of great help to improve our manuscript. According to your guidance, we have improved the revised manuscript point by point. And your comments were responded and explained in detail as follows.

To easily distinguish my answers from reviewer’s comments, I show my answer in red font while keeping your letter and reviewers’ comments in black font in this reply letter.

Point 1: The arrangement of sentence placement must be re-arranged, according to the existing template, because there is still a lot of empty space on the left side.

Response 1: Thank you very much for your suggestion. The arrangement of sentence placement is based on the format of published papers in this journal. We will confirm with the editor whether to leave the empty space on the left side, and then adjust the format.

Point 2: the purpose of this research for the contribution has not clear

Response 2: We apologize for not being clear about the purpose of the research. We supplement the research purpose in the abstract as follows: The research results provide a path for the reduction of carbon emissions and the improvement of eco-efficiency of Chinese tourist hotels. Under the backdrop of global climate change and the post-COVID-19 era, the research framework and conclusions provide references for countries with new economies similar to China and countries that need to quickly restore the hotel industry. Line 32-36.

Moreover, we also added the purpose and contribution in introduction parts as follows: From the aspect of theoretical and practical contribution, this study combed the spatio-temporal evolution pattern of income, carbon emissions, eco-efficiency and economic efficiency of Chinese tourist hotels in the past 20 years. The results aimed to narrow the spatio-temporal heterogeneity between the east and the west, give full play to the regional advantages of different regions, narrow the gap between the rich and the poor, and promote the coordinated development of China. Moreover, this study constructed the evaluation index system and theoretical framework for measuring eco-efficiency, provided the idea for measuring the relationship between economic development and the ecological environment, and regarded tourist hotels as an open eco-economic system interacting with the outside, thus constructing a two-stage regression model, to explore the factors that directly and indirectly affect eco-efficiency, and provides a path to optimize eco-efficiency for business operators and government entities. Line 101-112.

Point 3: the suggestions from the conclusions are also still hanging, please provide realistic suggestions so that business people can also get a direct impact, not just ". Regional governments should optimize the industrial structure of the hotel industry and improve the urbanization rate, energy efficiency, and information technology." for example: "improve the urbanization rate" do you have provided the data? is this still relevant to this research.

Response 3:Thank you very much for your advice. We are very sorry that the suggestions from the conclusions are hanging. We have added the realistic suggestions in the implications part, as follows: Line687-748.

The theoretical significance of this study is constructed a research framework from measurement to pattern analysis to driving mechanism, tried to find the scientific tools and paths for the green and low-carbon sustainable development of tourist hotels. This study uses the Super-SBM Non-Oriented model, constructs the index system and theo-retical framework for measuring eco-efficiency firstly, and reveals the spatio-temporal evolution pattern of the income, carbon emissions, eco-efficiency and economic efficiency of Chinese tourist hotels from 2000 to 2019. It is hoped that the research results can provide ideas for balancing economic development and the ecological environment, as well as suggestions for regional coordinated development in China. Secondly, through the CR model, the coupling relationship between income and carbon emission, eco-efficiency and economic efficiency is found. It is hoped that the research results can provide targeted transformation strategies according to the development characteristics of the hotel in-dustry in different provinces. Finally, this paper regards tourist hotels as a complete eco-economic system, and through a two-stage regression model, finds out the factors that directly and indirectly affect the eco-efficiency of tourist hotels, hoping that the research results can provide ideas for enterprises and governments to improve the eco-efficiency. Moreover, the following recommendations are made based on the results of this paper's empirical study.

Accelerate the transition of China's tourist hotels from a crude development model to a low-carbon, green and sustainable development model. The provinces with no ad-vantages in eco-efficiency should deepen the reform of the property rights system of the hotel industry based on their resource advantages, promote the diversification of the property rights structure of hotels, learn from the development experience of the hotel industry in Jiangsu Province, and realize the transformation from a low-carbon emission economic province to a high-carbon emission economic province as soon as possible. For example, in the production process, the geothermal system is used instead of gas heating, heating and air conditioning system that can be turned off automatically when guests leave the room to sleep, LED energy-saving lamps are used, and heat insulation window materials to prevent heat loss are used, and employees are encouraged to use environmentally friendly vehicles. In the consumption process, ceramic tableware is used instead of plastic tableware, and slogans with moral constraints are posted, such as "saving resources is everyone's responsibility" and "protecting the earth, refusing to waste". Long-term customers change bed sheets and toiletries as required instead of daily changes, collecting and purifying rainwater for watering flowers and raising fish, and encouraging consumers to use non-disposable products.

Enterprises and the government should jointly adjust the internal and external influencing factors of the tourist hotel system, and then adjust the eco-efficiency of tourist hotels. Within the tourist hotel system, enterprises can directly adjust the eco-efficiency of tourist hotels by adjusting the capital effect, scale effect, water consumption and carbon emission, such as limiting the proportion of technical investment in the initial investment and replacement investment in the hotel industry, using short videos to publicize and expand the market. Outside the tourist hotel system, enterprises should call on the government to adjust the local industrial structure, urbanization, energy efficiency and information degree as soon as possible to indirectly adjust the eco-efficiency of tourist hotels. For example, to expand the scale of hotel construction to increase the proportion of tourism and other tertiary industries, to promote exchanges between communities and tourist hotels to raise residents' awareness of environmental protection, to publicize the awareness of energy efficient use and regularly assess the results, to improve the degree of information construction and transportation convenience.

Point 4: overall your research is good and very interested  

Response 4:Thank you very much for your comments on this paper, it has greatly encouraged our research enthusiasm.

Reviewer 2 Report

Thank you for providing me with the opportunity to review your paper. I have enjoyed reading it. This is an interesting study. Overall, the paper is well-written and organized. However, I believe that further work is necessary before the paper is suitable for publication

I have the following observations:  

1. One major concern relates to the lack of academic strength of the paper. Although the authors provide a summary of the existing literature in the field, it remains unclear how exactly they extend this literature. More specifically, it is not very clear what contribution your paper makes to the extant literature.

2. The discussion is at a fairly general level. Some supplementary literature must be added to compare and contrast the key findings with the existing study. The discussion needs greater engagement with the literature to bring it up to an appropriate level.

3. In section 4., the authors must clarify the contributions to the literature. The conclusions of the study should be put first and only then should the implications of the study be stated. In addition to the practical implications, the authors should also mention the theoretical implications.

4. Recheck the references and their style are according to the journal requirements, and in-text and end-text should be the same and vice versa.

I hope my feedback on this paper will help the authors to improve the manuscript.

Author Response

Dear Reviewer,

Thank you very much for your professional and crucial comments and guidance, and we do think it is of great help to improve our manuscript. According to your guidance, we have improved the revised manuscript point by point. And your comments were responded and explained in detail as follows.

To easily distinguish my answers from reviewer’s comments, I show my answer in red font while keeping your letter and reviewers’ comments in black font in this reply letter.

Point 1: One major concern relates to the lack of academic strength of the paper. Although the authors provide a summary of the existing literature in the field, it remains unclear how exactly they extend this literature. More specifically, it is not very clear what contribution your paper makes to the extant literature. 

Response 1: Thank you very much for your guidance and suggestions. According to your suggestions, in order to increase the academic strength of the paper and the expansion of the existing research, we have added section 2 “literature review and research framework”, which summarizes the existing research (Line118-157 )and elaborates the theoretical basis (Line186-216 )of the research framework in this paper.

In addition, we also supplement the theoretical and practical contributions of this study in the introduction. As follows: Line101-112

From the aspect of theoretical and practical contribution, this study combed the spatio-temporal evolution pattern of income, carbon emissions, eco-efficiency and economic efficiency of Chinese tourist hotels in the past 20 years. The results aimed to narrow the spatio-temporal heterogeneity between the east and the west, give full play to the regional advantages of different regions, narrow the gap between the rich and the poor, and promote the coordinated development of China. Moreover, this study constructed the evaluation index system and theoretical framework for measuring eco-efficiency, provided the idea for measuring the relationship between economic development and the ecological environment, and regarded tourist hotels as an open eco-economic system interacting with the outside, thus constructing a two-stage regression model, to explore the factors that directly and indirectly affect eco-efficiency, and provides a path to optimize eco-efficiency for business operators and government entities.

Point 2: The discussion is at a fairly general level. Some supplementary literature must be added to compare and contrast the key findings with the existing study. The discussion needs greater engagement with the literature to bring it up to an appropriate level.

Response 2: Thank you very much for your guidance. According to your guidance, we separated the discussion from the original results section, and made a comparative analysis of the 11 relevant literatures and discussed the key conclusions. We hope the current discussion has been improved. The discussions as follows: Line616-656.

The focus of income and carbon emission of tourist hotels in China has shifted from the southeast to the northwest, which showed that the western provinces had not coordinated the relationship with the ecological environment while developing their economy rapidly. According to previous studies, economic development at the cost of the environment is not the best development path. It is suggested that the western region of China should adjust its development strategy in time and emphasize the irreversibility of environmental damage. In this study period, the eco-efficiency of Chinese tourist hotels is generally higher than the economic efficiency, which is different from the general law of regional eco-efficiency of Chinese provinces. Xia studied the regional economic efficiency and eco-efficiency of tourist hotel in China in 2016, found that the economic efficiency of each province is higher than the eco-efficiency level. According to the existing research, it is found that the eco-efficiency of regional tourism under the constraint of the ecological environment shows an obvious downwards trend, while a slight increase in the eco-efficiency of tourist hotels shows that, compared with other tourism sectors, Chinese tourist hotels have comparative advantages in balancing economic growth and ecological environmental impact. these findings will facilate that researchers focus on carbon reduction measures in western provinces.

From this study, it is found that the development of China's tourist hotels is still in the extensive development stage. Long found that the technical efficiency of China's tourist hotels is low when studying the space-time heterogeneity of the hotel industry's environmental efficiency. For example, operators refuse to use solar energy instead of electric energy because they are worried that unstable solar energy will be complained about by customers on rainy, cloudy and snowy days, thus affecting their operating income. Tang found that there are obvious phenomena of food waste and water waste in China's high-star hotels. Jie studied the low-carbon behavior of employees in star-rated hotels, found that the employees in star-rated hotels had insufficient low-carbon operation concept, which could not guide consumers well Combined with the re-search in this paper, it is found that if tourist hotels can't adjust their business philosophy and structure in time, they will still be unable to realize the green and low-carbon transformation for a long time to come.

The internal and external indicators of star-rated hotel system directly and indirectly, affect eco-efficiency. Moutinhoand and Robaina-Alvesstudied tourism satellite accounts in Portugal by using decomposition analysis. Both studies found that the tourism activity effect, energy over fixed capital, and capital over labor productivity all have significant impacts on the accommodation industry. When studying the eco-efficiency of tourism, it is found that energy technology is the key to improving the eco-efficiency, which is consistent with the internal research of star-rated hotel system in this paper, that is, excessive energy consumption and capital investment have a negative impact on the eco-efficiency of star-rated hotels. At the same time, according to the input-output process of star-rated hotels, some factors often do not directly affect the eco-efficiency of hotels. Li found that the optimization of industrial structure and urbanization rationalized the al-location of labor and energy consumption in the hotel industry, and the improvement of industrial structure would also improve the efficiency of CO2 emission. In addition, with the increase in the proportion of urban residents, hotels could reduce the idle labor force. This is the same as the research result of this paper. These findings are helpful for managers to know how to improve eco-efficiency.

Point 3: In section 4., the authors must clarify the contributions to the literature. The conclusions of the study should be put first and only then should the implications of the study be stated. In addition to the practical implications, the authors should also mention the theoretical implications. 

Response 3: Thank you very much for your guidance. We have reorganized the conclusions. Now it is conclusions and implications, and the implications is placed after the conclusions. At the same time, we further elaborated the theoretical and practical implications of this research, hoping that it can meet your requirements and suggestions at present.

The revised implications as follows: Line687-748

The theoretical significance of this study is constructed a research framework from measurement to pattern analysis to driving mechanism, tried to find the scientific tools and paths for the green and low-carbon sustainable development of tourist hotels. This study uses the Super-SBM Non-Oriented model, constructs the index system and theoretical framework for measuring eco-efficiency firstly, and reveals the spatio-temporal evolution pattern of the income, carbon emissions, eco-efficiency and economic efficiency of Chinese tourist hotels from 2000 to 2019. It is hoped that the research results can provide ideas for balancing economic development and the ecological environment, as well as suggestions for regional coordinated development in China. Secondly, through the CR model, the coupling relationship between income and carbon emission, eco-efficiency and economic efficiency is found. It is hoped that the research results can provide targeted transformation strategies according to the development characteristics of the hotel industry in different provinces. Finally, this paper regards tourist hotels as a complete eco-economic system, and through a two-stage regression model, finds out the factors that directly and indirectly affect the eco-efficiency of tourist hotels, hoping that the research results can provide ideas for enterprises and governments to improve the eco-efficiency. Moreover, the following recommendations are made based on the results of this paper's empirical study.

Consolidate and expand the advantages of the Yangtze River Delta region, strengthen the radiation of the Pearl River Delta region to the southwest region, and promote the linkage effect of the Beijing-Tianjin-Hebei region to the northeast and northwest regions. Provinces with insignificant spillover effects should take the initiative to help neighboring provinces and cities, and establish a cooperation mechanism for inter-provincial tourist hotels. For example, holding a sharing meeting of hotel management experience, establishing a sharing discount mechanism, and exchanging customer sources based on ensuring customer privacy. The provinces without development advantages should actively learn from the advanced technology and management experience of tourist hotels in neighboring provinces. For example, referring to the development history of excellent hotels and combining their actual conditions, they should formulate development strategies, ask relevant industry experts for technical guidance, and train their employees regularly.

Accelerate the transition of China's tourist hotels from a crude development model to a low-carbon, green and sustainable development model. The provinces with no ad-vantages in eco-efficiency should deepen the reform of the property rights system of the hotel industry based on their resource advantages, promote the diversification of the property rights structure of hotels, learn from the development experience of the hotel industry in Jiangsu Province, and realize the transformation from a low-carbon emission economic province to a high-carbon emission economic province as soon as possible. For example, in the production process, the geothermal system is used instead of gas heating, heating and air conditioning system that can be turned off automatically when guests leave the room to sleep, LED energy-saving lamps are used, and heat insulation window materials to prevent heat loss are used, and employees are encouraged to use environmentally friendly vehicles. In the consumption process, ceramic tableware is used instead of plastic tableware, and slogans with moral constraints are posted, such as "saving resources is everyone's responsibility" and "protecting the earth, refusing to waste". Long-term customers change bed sheets and toiletries as required instead of daily changes, collecting and purifying rainwater for watering flowers and raising fish, and encouraging consumers to use non-disposable products.

Enterprises and the government should jointly adjust the internal and external influencing factors of the tourist hotel system, and then adjust the eco-efficiency of tourist hotels. Within the tourist hotel system, enterprises can directly adjust the eco-efficiency of tourist hotels by adjusting the capital effect, scale effect, water consumption and carbon emission, such as limiting the proportion of technical investment in the initial investment and replacement investment in the hotel industry, using short videos to publicize and expand the market. Outside the tourist hotel system, enterprises should call on the government to adjust the local industrial structure, urbanization, energy efficiency and information degree as soon as possible to indirectly adjust the eco-efficiency of tourist hotels. For example, to expand the scale of hotel construction to increase the proportion of tourism and other tertiary industries, to promote exchanges between communities and tourist hotels to raise residents' awareness of environmental protection, to publicize the awareness of energy efficient use and regularly assess the results, to improve the degree of information construction and transportation convenience.

Point 4: Recheck the references and their style are according to the journal requirements, and in-text and end-text should be the same and vice versa.

Response 4: Thank you very much for your suggestion. We have sorted out the references in-text and end-text according to the ACS format which required from Author Guide of the journal.

I hope my feedback on this paper will help the authors to improve the manuscript.

Response: Thank you very much for your comments on this paper, it has greatly helped to improve our manuscript.

Round 2

Reviewer 2 Report

All my suggestions have been properly responded.

Recommendation: Accept